# Analysis of the Relationship between Corporate CSR Investment and Business Performance Using ESG Index—The Use-Case of Korean Companies

**Jihye Yun [1] and Jonghwa Lee [2],***

[1]   Business Administration, Kyung-sung University, 309, Suyeong-ro, Nam-gu, Busan 48434, Korea; jhys01@hanmail.net
[2]   Department of e-Business, Dong-eui University, 176, Eomgwang-ro, Busanjin-gu, Busan 47340, Korea
*   Correspondence: jhlee6050@deu.ac.kr; Tel.: +82-10-3526-6050

**Abstract:** ESG management has become the most prominent topic this year as the worldwide business environment of companies changes. ESG management is not an option to raise a company's reputation but a factor that significantly affects corporate sustainability and corporate value in the long run as well as can be seen as an essential requirement for a company's survival. As this trend is essential in evaluating corporate sustainability, ESG management is an inevitable global issue. This study aims to develop an index for ESG, a key indicator of corporate sustainability, targeting 49 companies that issue 'sustainability reports' among the top 150 asset-ranked companies. In addition, companies were divided into high and low groups based on the ESG index, and each group attempted to study the effect of economic responsibility among corporate social responsibility on financial performance. As a result, out of 49 companies, the high group was divided into 17 companies, and the low group was split into 14 companies. It was found that only the low group had a significantly positive (+) effect on the relationship between corporate economic responsibility and financial performance. This study can be used as a reference data for ESG research by developing an ESG index related to corporate sustainability. It is meaningful to confirm the management paradigm for corporate social responsibility and the importance of financial performance through comparison.

**Keywords:** ESG; CSR; business performance; sustainability report; non-negative matrix factorization

## 1. Introduction

Recently, developed countries are in the process of promoting various environmental regulations to respond to climate change, and ESG management is becoming a widely discussed topic worldwide. ESG management, which has established itself as a new standard in corporate management, includes carbon reduction, social contribution, circular economy, and corporate governance. In recent years, Korean companies have rapidly adopted ESG management to keep up with the demands of the new era, recognizing sustainability management as one of their active management strategies; they are striving to fulfill the corporate social responsibility by protecting customer information and the environment in consideration of various stakeholders and acknowledging the diversity of our employees [1]. Furthermore, consumers have started to evaluate the company based on the value that the company pursues, not simply based on the products or services. This trend lets ESG emerge and makes it an essential element in assessing the sustainability of corporate [2]. As ESG became the core of corporate management, international organizations such as the European Commission (EC), the International Organization of Securities Commissions (IOSCO), and the OECD began to understand the current status and problems of ESG evaluation. In addition, the requirements for ESG evaluation agencies to strengthen their transparency are increasing [3].

Amazon, which has grown into a global conglomerate with an aspiration to sell everything globally, has become a significant culprit in carbon emissions due to high power consumption and environmental pollution related to packaging and transportation in the process of rapid growth. This led them to take attention to ESG. Amazon founder and former CEO Jeff Bezos announced the subject of the "climate pledge" in September 2019, Ref. [4] and Amazon began to invest USD 2 billion won in clean energy-related companies to implement the climate pledge; it then began to acquire the aspects of a sustainable company by investing in a large-scale fund. In addition, it decided to use 100% renewable energy by 2030 and signed a contract in June 2021 to procure eco-friendly energy at 14 solar and wind power plants in the U.S. and Europe. Amazon is also cooperating with all its partners to achieve its goal of "zero" carbon emissions by 2040 [4]. By working from various aspects to treat environmental issues, Amazon is transforming into an eco-friendly company, and its reputation is growing as the company that uses the largest amount of renewable energy in the world. This trend is seen not only in Amazon, but in a lot of multinational companies such as Apple, which achieved 100% renewable energy, and Microsoft, which vowed carbon negative, have conducted sustainable management through successful ESG with their superior technologies.

POSCO, which published Korea's first "environmental report" in 1995, has jointly carried out improvement tasks with business partners since 2004, and then implemented the "performance sharing system" to share results [5]. In addition, when Chairman Choi Jung-woo took office in 2018, he emphasized the company's role as a "corporate citizen" who fulfills social responsibility, not just pursuing profits. The company declared "2050 Carbon Neutral" replacing energy sources for steel production with hydrogen instead of coal. This new policy included the new introduction of safe products and solutions that can reduce carbon instead of existing buildings, wood, stone, and aluminum for the first time in Korea, in December 2020, Ref. [5]. As such, POSCO is striving to respond to global climate change by expanding renewable energy in the medium and long term and actively practicing ESG management. In addition, several Korean manufacturing companies are trying to introduce ESG successfully: Samsung Electronics has expanded ESG management to its partners; SK Group expanded the use of renewable energy; LG Electronics included the environmental burden financially by introducing an internal carbon tax, and Hyundai Motor have planned to make large-scale investments on ESG by 2030.

Social contribution in ESG refers to corporate charity activities, and the Global Reporting Initiative (GRI) standard refers to the performance of social contribution activities as "GRI413-Local Communities" to disclose detailed activities such as social contribution strategies, expenditures, and performance [6]. In Korea, the annual social value report is published every December by the Federation of Korean Industries; they defined social contribution as companies' educational, cultural preservation activities, job creation, and social investment through support for the vulnerable. According to an analysis of 220 top Korean companies, social contribution expenditure was about USD 3 trillion won and average expenditure of one company was USD 13.6 billion won; indeed, the total expenditure has increased 14.8% from the previous year [7]. The above statistic clearly shows that the companies are increasingly interested in realizing social values through investment in social responsibility and environmental responsibility through ESG management, and they are trying out various ways to achieve shared growth with partners [8].

Recently, ESG research has studied the relationship between the business performance of companies only with rating results announced by evaluation agencies; however, it is not clear whether the company's ESG activities are reflected in the evaluation. Additionally, the validity and certainty of the evaluation items are not proven delicately. To fill this gap, this study aims to develop the ESG index, a key indicator for examining ESG from a non-financial perspective and evaluating corporate sustainability, based on existing ESG-related research and reports. In addition, we classified the companies into high and low groups according to the ESG index. The impact of CSR investment on management performance was also examined from a financial perspective. This study would be able to play a role

as the primary source of data for ESG-related research by developing an ESG index based on corporate sustainability. At the same time, this study has its implication in that we expanded the research field by reinterpreting the relationship between investment and management performance from a financial perspective through comparison of high and low groups; thus, this article can provide some proofs that the management paradigm of corporate social responsibility is now changing, and not only financial but non-financial perspectives also should be highlighted.

This study has been divided into five parts: the Section 1 described the background and purpose of this study. The Section 2 examines the concept and status of ESG from a non-financial point of view and ESG from a financial point of view based on previous studies and literature. The Section 3 presents the framework and three methodologies for ESG index development and establishes a hypothesis on the relationship between CSR investment and business performance. The Section 4 describes the relationship between the results of the ESG index development and the management performance of each group. Section 5 deals with the implications of this study and future research directions.

## 2. Theorical Background

### 2.1. Non-Financial ESG

Sustainable management is a management paradigm that pursues sustainable development based on social, economic, and environmental responsibilities to sustain corporate management more stably. It aims to construct "environment" and "governance" that does not harm society. Most of the companies' ultimate goal had been profit-seeking through technological innovation. Now, the safety and protection of humanity are the top priorities, giving more attention to technologies that solve social and environmental problems. As such, companies are rapidly changing to ESG management to value society, the environment, and people. In other words, ESG has established itself as a global standard for creating and practicing socially positive values beyond just generating profits [9].

In Europe, public announcement on ESG has been mandatory for all financial companies since March 2021, and investment activities are suspended under some conditions. Some Asian countries also encourage companies to participate in ESG management by providing related costs to ESG bond issuers. BlackRock, the world's largest asset management company, urges investment target companies to achieve zero carbon dioxide net emissions by 2050. Credit rating agencies Moody's, Peach, and S&P have decided to actively consider ESG-related factors when evaluating companies [10]; indeed, ESG directly affects companies in Korea as well. Korean Financial Services Commission recently mandated the issuance of sustainability reports by 2025 for listed companies with assets of 2 trillion won or more [11], and by 2030, all listed companies are obligated to disclose sustainability reports. In addition, Korean National Pension Service, one of the world's top three pension funds, said it would focus on managing companies with ESG ratings falling by more than two levels in April 2021. They recently decided to strengthen the investment on ESG by assigning weights on climate-related and industrial safety-related evaluation items. This decision shows that the companies should participate in the ESG much more actively than before. As ESG information disclosure becomes mandatory around the world and carbon reduction regulations are tightened, ESG management is expected to positively impact society and companies in the mid to long term, resulting in the growing demands of stakeholders on ESG.

Despite the increasing importance of ESG, grading systems and evaluation categories differ by organization. Evaluation systems have similar frameworks in terms of how to give additional points and deductions, but differ in detailed score calculation and weighting. Table 1 shows the ESG rating gap of Korean companies based on the evaluation of the MSCI (Morgan Stanley Capital International) in 2020. 22 companies with a rating gap of three or more stages among MSCI, Refinitiv, and Korean Corporate Governance Service (KOGS) are listed. MSCI's ESG evaluation uses publicly disclosed data, and the assessment ranges from AAA to CCC. They evaluate corporates yearly with 37 categories related to

environmental, social, and governance issues. Refinitiv also uses publicly available data and uses 0–100 point evaluation, evaluating the ESG Score and ESG Controversy Score with 10 items and update the score every two weeks. KOGS also evaluates companies yearly using publicly disclosed data, assessing environment, society, and governance sectors. They adopt 7-point scale: S, A+, A, B+, B, C, and D [12]. Considering the different scales of the above three ratings, we converted the scores of Refinitiv and KCGS into the scale of MSCI. The average ESG grade gap ranges from at least 1.4 stages to 5 stages [13]. Those gaps among assessments well show the inconsistency of ESG ratings, emphasizing the necessity of an objective ESG index.

**Table 1.** Comparison results of ESG rating gap by company based on MSCI.

| Company Name | Adjustment Level * | | | Rating Gap | | | |
|---|---|---|---|---|---|---|---|
| | MSCI (Level 7) | Refinitiv (Out of 100 Points) | KCGS (Level 7) | M-R ** | M-K ** | R-K ** | Average Gap |
| Hyundai Steel | CCC | AA (77) | BBB (B+) | 5 | 3 | 2 | |
| Kia Motors | CCC | A (62) | A (A) | 4 | 4 | 0 | |
| Hyundai Motor | B | AA (74) | A (A) | 4 | 3 | 1 | |
| Samsung Heavy Industries | CCC | A (64) | BBB (B+) | 4 | 3 | 1 | |
| Korea Electric Power Corporation Co., Ltd. | BB | AA (73) | A (A) | 3 | 2 | 1 | |
| Korea Gas Corporation Co., Ltd. | BB | AA (73) | A (A) | 3 | 2 | 1 | |
| Hyundai Glovis Co., Ltd. | BB | AA (77) | A (A) | 3 | 2 | 1 | |
| Hyundai Engineering & Construction Co., Ltd. | BB | AA (81) | A (A) | 3 | 2 | 1 | |
| Doosan Heavy Industries & Construction Co., Ltd. | BB | AA (74) | A (A) | 3 | 2 | 1 | |
| S-Oil Co., Ltd. | BB | AA (82) | AA (A+) | 3 | 3 | 0 | 2.2 |
| Hyundai Mobis Co., Ltd. | B | BBB (51) | A (A) | 2 | 3 | 1 | |
| Lotte Shopping Co., Ltd. | B | BBB (49) | A (A) | 2 | 3 | 1 | |
| E-Mart | B | BB (36) | A (A) | 1 | 3 | 2 | |
| Kumho Petrochemical Co., Ltd. | B | B (27) | A (A) | 0 | 3 | 3 | |
| BGF Retail | BB | CCC (12) | A (A) | 2 | 2 | 4 | |
| S1 Corporation | BB | CCC (9) | BBB (B+) | 2 | 1 | 3 | |
| CJ Logistics Co., Ltd. | BB | B (20) | A (A) | 1 | 2 | 3 | |
| The Shilla Hotels & Resorts | BB | B (21) | A (A) | 1 | 2 | 3 | |
| Korea Aerospace Industries Co., Ltd. | BB | B (22) | A (A) | 1 | 2 | 3 | |
| Ottogi Corporation Co., Ltd. | B | CCC (8) | BBB (B+) | 1 | 2 | 3 | |
| Samsung Electronics Co., Ltd. | A | AAA (91) | BBB (B+) | 2 | 1 | 3 | |
| LG Electronics Co., Ltd. | A | AAA (90) | BBB (B+) | 2 | 1 | 3 | |

* Rating system: (MSCI) AAA, AA, A, BBB, BB, B, CCC, (KCGS) S, A+, A, B+, B, C, D. (Refinitiv) Converts the 100-point scoring system into 7 level grades at 14-point intervals. ** The abbreviation is M (MSCI), R (Refinitiv), K (KCGS), which means the difference in grades between the evaluation grades of each institution. Source: The Federation of Korean Industries. Domestic and Overseas ESG Evaluation Trends and Implications. ESG MEMO, Vol. 2, 1–3.

As each institution provides and utilizes different ESG evaluation results, the benchmark indicator is needed for presenting clear directions to each company. Companies should not confuse ESG with CSR or CSV activities, but should approach the core of management strategies to achieve sustainable growth.

As such, ESG has a different financial situation for each company and different performance areas due to differences in technology, so it is not easy to regulate or evaluate it in a uniform framework. Strengthening ESG management can make it difficult for companies

with poor business conditions to maintain their survival. Therefore, governmental support is needed, while easing the legal and emotional regulations so that companies can participate in ESG autonomously. In addition, evaluation items for ESG activities of companies should be diversified, and evaluation contents should be standardized by company size so that the evaluation gap between companies by the institution can be narrowed. As a result of a recent study using the company's ESG evaluation as a proxy for the circular economy (CE) concept, it was found that ESG evaluation has a positive effect on the stock return ratio; however, it was argued that a higher ESG score in itself does not increase the stock price [14]. Therefore, ESG positively impacts the environment, society, and economy worldwide.

Therefore, to diversify the evaluation of ESG, this study intends to develop the ESG index by weighting ESG keywords using the results of previous studies by researchers who extracted ESG keywords through "sustainability reports" by the company.

### 2.2. Financial Perspective ESG

As ESG emerged as a new paradigm in corporate management, "ESG" is 1.2 times more mentioned in 2020 than "Social Contribution" in Google; however, as of February 2021, it was 7.5 times more mentioned about "ESG" than "Social Contribution" [6]. This trend shows that companies' social contribution spending is steadily expanding to the whole field of ESG, not limited to the social contribution.

Carroll (1991) divided corporate responsibility into economic, legal, ethical, and charitable responsibilities by presenting a Pyramid of Corporate Social Responsibility [15]. The economic responsibility of companies, located in the bottom of the pyramid, is to produce goods and services as the primary economic unit of society. Jobs are created through production, distribution, and sales, and employees can earn a living. In addition, consumers can improve their quality of life by purchasing new products. Thus, corporate economic responsibility becomes the basis of all other responsibilities. Corporate management is responsible for complying with the law and implementing it fairly and justly, which is both legal and ethical responsibility. Charitable responsibility is based on economic, legal, and ethical responsibilities. From a financial perspective, it can be seen as ESG, including economic responsibility, to become a good corporate citizen and provide community resources through corporate social contribution. Corporate social contribution activities refer to a voluntary contribution to social development by utilizing valuable resources created by the efforts of corporate members as members of the community and can be seen as a promise to society [16]. Therefore, companies should pay attention to social issues and solutions to create a better local community and strengthen social contribution activities through partnerships between the government and civic groups.

According to a survey conducted by the Federation of Korean Industries on social contribution activities, social contribution expenditures have changed, as shown in Figure 1. Figure 2 shows the fluctuation of social contribution expenditure. 38.8% of companies experienced a decrease in social contribution expenditures in 2019 compared to 2018. On the contrary, 54.3% of them increased their social contribution expenditure in 2019. The fluctuation in social contribution expenditure happened because each company is engaged in social contribution activities in consideration of policies and social issues [7]. As of 2019, 23.9% of corporate activities focused on realizing social values with partners, followed by 20.9% on eco-friendly values, and 20.9% on strengthening compliance management. Companies strive to engage in various social contribution activities by securing transparency in fair trade and strengthening competition among partners for shared growth with partners. Social contribution does not stop at temporarily solving social problems, but includes all non-profit actions to solve social problems at the local community level and improve the quality of life. It is divided into physical and human resources: physical contribution activities refer to the form in which companies directly donate, and human resources mainly include volunteer activities [17]. Since a company's fundamental purpose is to pursue profits, social contribution activities should be carried out in consideration of

the company's interests or survival. If possible, it should focus on a direction that helps increase its profits [18]. So far, many studies have shown that companies must use charitable donations as part of their strategic activities to gain a competitive position and that strategic donation activities ultimately help companies' management performance [19,20].

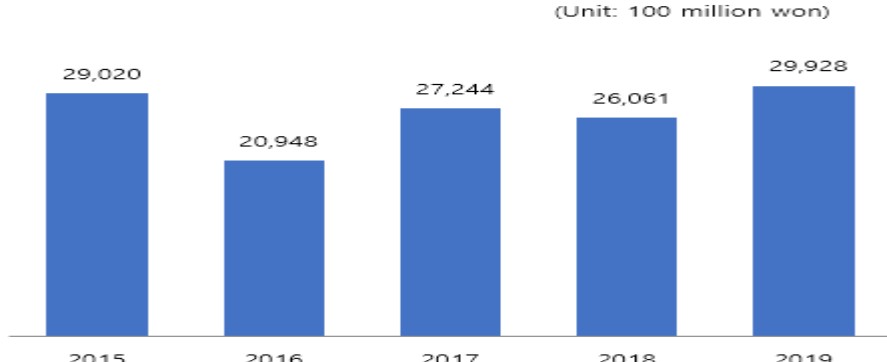

**Figure 1.** The size of corporate social contribution expenditure over the past five years. Source: The Federation of Korean Industries, Message Report from Major Companies in 2020.

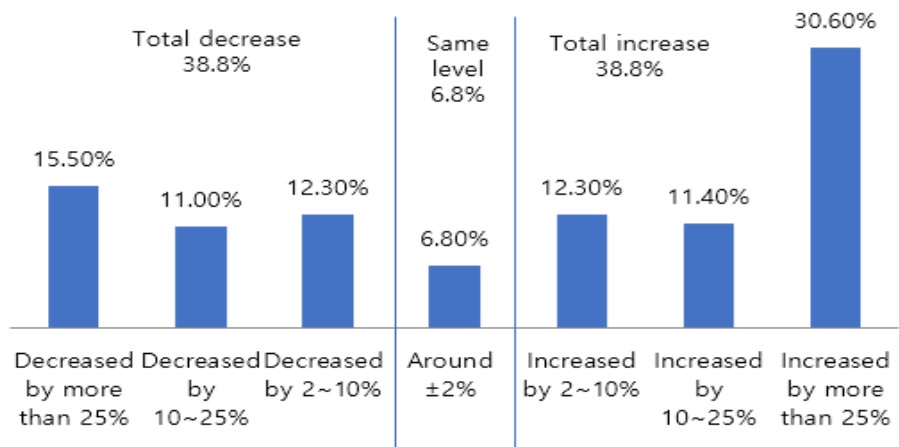

**Figure 2.** Distribution of social contribution fluctuation in 2018–2019. Source: The Federation of Korean Industries, Message Report from Major Companies in 2020.

Considering the previous studies related to ESG from a financial point of view, Mario and his colleges analyzed panel data on the relationship between ESG investment and stock returns for 46 companies using data from 2010 to 2018. As a result of the analysis, it was found that ESG investment and business performance are related in the short-term but not in the long term [21]; moreover, there have been an increasing number of studies analyzing sustainability from various perspectives in recent years. The study of Kovacova investigated whether an automated production system and industrial artificial intelligence based on big data helps construct organizational performance [22]. In addition, Dawson analyzed the business management system and performance to confirm the relationship between sustainable product lifecycle management and organizational performance [23]. Lăzăroiu performed an analysis of the cyber-physical production network of a sustainable manufacturing system and analyzed the relationship with the company's performance through data [24]. As far, ESG is closely related to companies such as production, management, investment, and performance, and as a result, research topics on ESG and management performance have been diversifying.

Therefore, this study attempts to analyze whether there is a significant relationship between social contribution activities and management performance, which are relatively unclear and ambiguous concepts by using the ESG index. Donations, which can be seen as

a relatively objective criterion for social distribution, are regarded as financial ESG. This study aims to see whether donations have a significant effect on a company's management performance.

## 3. Research Methods and Design

### 3.1. ESG Index Development

One of the management indicators was to reduce the defect rate in the production process, avoid injuries to workers, and achieve the goal of products produced in the traditional manufacturing-based industry. In terms of business performance, maximizing efficiency from an economic point of view was the company's survival strategy; however, as ICT develops, industrial efficiency from a mere economic standpoint alone cannot preempt the competition and cannot protect the company from the threat of suppliers, competitors, and customers. Thus, the importance of environmentally responsible management, socially responsible management, and transparent management is growing and emphasized, which is the basic framework of ESG management [25]. Since economic and social values occur together, corporate actions interact with the benefit of society, and responsible social activities are becoming the driving force of corporations.

In addition, ESG management is becoming a globally emerging issue in terms of its role as an important factor in assessing companies' sustainability [26]. The study aims to develop the ESG index evaluating sustainability with 49 companies issuing a sustainability report among the top 150 Korean companies. ESG keywords were extracted from the CEO messages in Sustainability Report from 2013 to 2020, and those were used to calculate the ESG index.

Figure 3 is a framework for the development of ESG indices. Using the CEO's greeting from the report as the raw material, we calculated weights using the TF-IDF technique compared to the ESG keyword, the result of the researcher's previous study. TF-IDF is a more reliable technique than relying on simple frequency as a weight representing how important a word is within a particular document using the frequency of appearance. In other words, it is a technique obtained by multiplying the frequency and frequency of reverse documents with keywords that appear in a large number of documents [27]. In addition, the characteristics of ESG words were extracted using NMF (Non-Negative Matrix Factorization), a document classification technique. NMF is a technique for decomposing the extracted characteristic matrix into two matrices, W (weighted) and H (characteristic) [28]. It has the advantage of being quickly decomposed by using features that do not have negative elements in the data. The extraction method for the NMF technique was carried out with three methodologies.

The extraction method for the NMF technique was carried out with three main techniques. Data collection was downloaded directly from each company's website. Next, text mining was analyzed using Linux_R. In addition, we developed algorithms using R packages tm and NMFN.

(1)    NMF_mm multiplicative update method

As a basic method of the NMF technique, it is a technique applied using the product of the matrix (Equation (1)). The matrix is updated by Equations (2) and (3) and is repeated until satisfied.

$$V = WH \tag{1}$$

$$H_{\alpha\mu} = H_{\alpha\mu} \frac{(W^T A)_{\alpha\mu}}{(W^T WH)_{\alpha\mu}} \tag{2}$$

$$W_{ia} = W_{ia} \frac{(AH^T)_{ia}}{(WHH^T)_{ia}} \tag{3}$$

(2)    NMF_als—alternating least-squares method

$$\min_{x,y} \sum_{u,i} c_{ui}(p_{ui} - x_u^T y_i - \beta_u - \beta_i)^2 \\ +\lambda(\sum_u \| x_u \|^2 + \sum_i \| y_i \|^2) \tag{4}$$

Alternating least-squares involves calculating a single feature vector using a least-squares function (Equation (4)) in a way that repeatedly calculates and obtains a solution, and normalization is applied to prevent overfitting in the process of minimizing errors. In order to calculate the user feature vector until the optimal solution is obtained, it refers to a method of first fixing the item feature vector as a constant and solving it with the least number of squares.

(3)    NMF_prob—multinomial method

The polynomial method multiplies the weight (Equation (5)) and the characteristic (Equation (6)) using a diagonal matrix and is calculated by applying the following equation.

$$W = W * \sqrt{P} * \sum x \tag{5}$$

$$H = \sqrt{P} * H \tag{6}$$

$$P = \begin{bmatrix} \frac{1}{k} & & \\ & \ddots & \\ & & \frac{1}{k} \end{bmatrix}$$

When using the diagonal matrix of P, the weight (W) consists of a product of itself, a diagonal matrix, and the number of factors ($k$), and the characteristic (H) consists of a diagonal matrix and its product, calculated as the matrix product of Equation (1).

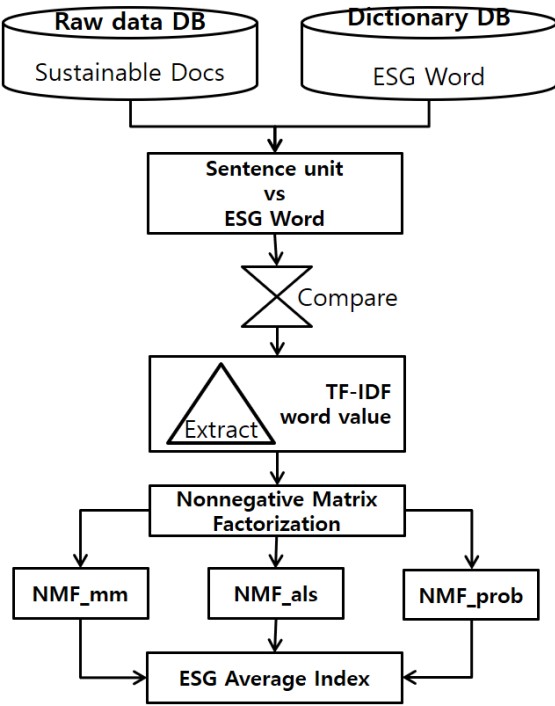

**Figure 3.** ESG Index Development Framework.

*3.2. CSR Investment and Business Performance*

A financial statement containing financial information show the company's current status, and the company's financial performance can be confirmed by increasing sales and profits and rising stock prices. In addition, investors mainly focused on financial factors when measuring corporate value before the importance of non-financial factors

emerged recently; thus, corporate value has been evaluated by quantitative and short-term indicators such as financial statements. In addition, traditionally, the size of management compensation has been determined based on market value and profitability, and accounting information in financial statements has been mainly used when determining compensation by measuring management performance [29]. Among them, donations in financial statements are voluntary expenditures by companies regardless of business activities and mean that they are used for the public interest, such as education, politics, culture, and social welfare. Indicators used in management performance include net profit per share, which indicates market performance; net income, which indicates profitability; sales and operating profit, and returns on assets. Therefore, in this study, donations are set as CSR investments, and the following hypothesis is established to empirically confirm the relationship between CSR investment and management performance by dividing the company's management performance by ROA (Return On Assets; ROA), Return on Equity (Return On Equity; ROE), and Return on Sales (Return On Sales; ROS).

**Hypothesis 1.** *CSR investment will have a significant positive (+) effect on a company's management performance.*

In order to correctly estimate the effect of CSR investment on the management performance of a company, other exogenous variables are controlled. In this study, the company's size, the presence or absence of a chaebol, ownership type, and advertising cost were considered control variables. In addition, CSR investment donations were divided into total assets to confirm the ratio of donations for the year, and the company's management performance was set as the year (t) and the following year (t + 1) to secure causality and confirm the Time-lag effect.

## 4. Research Results

In order to develop the ESG index, weights were extracted using the TF-IDF technique, and after extracting the characteristic values of words using NMF, the formula of the research method was used. The ESG index of 339 data of 49 companies was calculated from 2013 to 2020, and the ESG index of each company was confirmed by applying three indices. Method (1) NMF_mm used for calculation will be described by naming NMF_mm ESG Index 1, Method (2) NMF_als ESG Index 2, Method (3) NMF_prob ESG Index 3. The minimum value of ESG index 1 was 0.362, the maximum value was 6.594, and the standard deviation was 1.060. The minimum value of ESG index 2 was 4.756, the maximum value was 31.609, and the standard deviation was 4.821. Finally, the minimum value of ESG index 3 was 7.253, and the maximum value was 129.627, and the standard deviation was 21.229. In addition, the average value for each index was calculated using the developed ESG index, and ESG index 1 was 3.748, ESG index 2 was 18.980, and ESG index 3 was 53.542, whose average was 25.578. Samsung Heavy Industries had the lowest ESG index at 13.536. Lotte Shopping had the highest ESG index, followed by POSCO Energy with 36.928.

In addition, to empirically analyze the relationship between CSR investment and management performance, this study used 339 data from 2013 to 2020 and 290 data from t + 1 for 49 companies that published sustainability reports. Based on the average values of ESG indices 1, 2, and 3, all 17 companies that appeared above average were divided into stubborn groups and 14 companies that appeared below average were divided into low groups. The descriptive statistics and correlation coefficients of the variables used in the empirical analysis are presented in Table 2 (high group) and Table 3 (low group). The relationship between CSR investment and management performance in this study is consistent with the hypothesis predicted.

**Table 2.** Descriptive statistics and correlation of high group.

|  | Mean | S.D | 1 | 2 | 3 | 4 | 5 | 6 | 7 | 8 | 9 | 10 | 11 |
|---|---|---|---|---|---|---|---|---|---|---|---|---|---|
| 1. Scale | 30.737 | 1.350 | 1 | | | | | | | | | | |
| 2. Chaebol | 0.550 | 0.500 | 0.052 | 1 | | | | | | | | | |
| 3. Ownership | 1.860 | 0.348 | −0.197 * | 0.039 | 1 | | | | | | | | |
| 4. Advertising ratio | 0.577 | 0.012 | −0.074 | −0.115 | 0.026 | 1 | | | | | | | |
| 5. CSR investment | 0.105 | 0.132 | −0.307 ** | −0.146 | 0.151 | 0.226 * | 1 | | | | | | |
| 6. ROA (year t) | 2.735 | 4.774 | 0.147 | 0.351 ** | 0.285 ** | 0.302 ** | 0.069 | 1 | | | | | |
| 7. ROE (year t) | 5.513 | 14.504 | −0.030 | 0.300 ** | 0.258 ** | 0.096 | 0.030 | 0.857 ** | 1 | | | | |
| 8. ROS (year t) | 6.718 | 6.884 | −0.030 | −0.099 | 0.047 | −0.037 | 0.042 | 0.126 | 0.095 | 1 | | | |
| 9. ROA (year t + 1) | 2.578 | 4.816 | 0.123 | 0.357 ** | 0.305 ** | 0.215 * | 0.051 | 0.568 ** | 0.497 ** | 0.125 | 1 | | |
| 10. ROE (year t + 1) | 5.203 | 15.410 | −0.049 | 0.286 ** | 0.270 ** | 0.058 | 0.032 | 0.323 ** | 0.384 ** | 0.137 | 0.872 ** | 1 | |
| 11. ROS (year t + 1) | 7.819 | 7.457 | −0.054 | −0.109 | 0.045 | −0.023 | −0.044 | 0.077 | 0.080 | −0.010 | 0.138 | 0.099 | 1 |

* $p < 0.05$, ** $p < 0.01$.

**Table 3.** Descriptive statistics and correlation of low group.

|  | Mean | S.D | 1 | 2 | 3 | 4 | 5 | 6 | 7 | 8 | 9 | 10 | 11 |
|---|---|---|---|---|---|---|---|---|---|---|---|---|---|
| 1. Scale | 29.830 | 1.156 | 1 | | | | | | | | | | |
| 2. Chaebol | 0.720 | 0.453 | 0.075 | 1 | | | | | | | | | |
| 3. Ownership | 1.810 | 0.391 | −0.125 | 0.424 ** | 1 | | | | | | | | |
| 4. Advertising ratio | 0.012 | 0.017 | −0.137 | 0.087 | 0.229 * | 1 | | | | | | | |
| 5. CSR investment | 0.151 | 0.197 | −0.194 | −0.048 | −0.309 ** | 0.306 ** | 1 | | | | | | |
| 6. ROA(year t) | 4.186 | 5.880 | −0.044 | 0.191 | 0.150 | 0.405 ** | 0.430 ** | 1 | | | | | |
| 7. ROE(year t) | 6.903 | 11.787 | −0.124 | 0.195 * | 0.094 | 0.371 ** | 0.332 ** | 0.900 ** | 1 | | | | |
| 8. ROS(year t) | 5.590 | 9.143 | 0.011 | 0.147 | 0.168 | 0.192 | 0.267 ** | 0.845 ** | 0.763 ** | 1 | | | |
| 9. ROA (year t + 1) | 4.423 | 6.012 | −0.081 | 0.189 | 0.131 | 0.418 ** | 0.406 ** | 0.647 ** | 0.509 ** | 0.497 ** | 1 | | |
| 10. ROE (year t + 1) | 7.358 | 11.912 | −0.144 | 0.161 | 0.058 | 0.363 ** | 0.326 ** | 0.423 ** | 0.327 ** | 0.308 ** | 0.896 ** | 1 | |
| 11. ROS (year t + 1) | 5.968 | 9.474 | −0.031 | 0.143 | 0.154 | 0.185 | 0.319 ** | 0.531 ** | 0.418 ** | 0.442 ** | 0.833 ** | 0.754 ** | 1 |

* $p < 0.05$, ** $p < 0.01$.

The high group showed insignificant results in the correlation analysis, and the low group showed a significant correlation in the relationship between CSR investment and business performance. Thus, only the low group supported Hypothesis 1 through correlation analysis. Additional testing was performed using OLS linear regression analysis.

Table 4 presents a result of regression analysis with high group. CSR investments in companies in financial performance did not have a significant influence on ROA, ROE, ROS both at t year and t + 1 year. Table 5 shows the result of regression analysis with the low group. The relationship of CSR investment and management performance was confirmed. As for year t, ROA (β = 0.447, $p < 0.001$), ROE (β = 0.264, $p < 0.05$), and ROS (β = 0.367, $p < 0.01$) were all significantly influenced by CSR investment significantly positively (+). F value of regression equation with ROA as dependent variable was 9.409 ($p < 0.001$), and the modified $R^2$ was 0.294. F value of model with ROE was 5.638 ($p < 0.01$) and the modified $R^2$ was 0.187. F value with ROS model is 3.560 ($p < 0.001$) and the modified $R^2$ was 0.112.

**Table 4.** Results of regression analysis of the high group and business performance.

| Division | | Dependent Variable: Year t | | | Dependent Variable: Year t + 1 | | |
|---|---|---|---|---|---|---|---|
| | | ROA | ROE | ROS | ROA | ROE | ROS |
| Control variable | Company size | 0.234 ** | 0.016 | −0.012 | 0.217 * | 0.009 | −0.068 |
| | Chaebol presence | 0.377 *** | 0.306 ** | −0.103 | 0.377 *** | 0.288 ** | −0.126 |
| | Type of possession | 0.296 *** | 0.244 ** | 0.040 | 0.318 ** | 0.254 * | 0.050 |
| | Ad rate | 0.338 *** | 0.122 | −0.058 | 0.244 ** | 0.078 | −0.022 |
| Independent variable | CSR Investment | 0.075 | 0.015 | 0.030 | 0.079 | 0.030 | −0.088 |
| F value | | | 4.347 ** | 0.346 | 8.458 *** | 3.410 ** | 0.454 |
| $R^2$ | | | 0.166 | 0.016 | 0.315 | 0.156 | 0.024 |
| Modified $R^2$ | | | 0.128 | −0.030 | 0.278 | 0.111 | −0.029 |

The regression coefficients are standardized. * $p < 0.05$, ** $p < 0.01$, *** $p < 0.001$.

**Table 5.** Results of regression analysis of the low group and business performance.

| Division | | Dependent Variable: Year t | | | Dependent Variable: Year t + 1 | | |
|---|---|---|---|---|---|---|---|
| | | ROA | ROE | ROS | ROA | ROE | ROS |
| Control variable | Company size | 0.092 | −0.045 | 0.118 | 0.039 | −0.074 | 0.084 |
| | Chaebol presence | 0.099 | 0.173 | 0.037 | 0.116 | 0.153 | 0.039 |
| | Type of possession | 0.206 | 0.037 | 0.273 * | 0.167 | 0.008 | 0.306 * |
| | Ad rate | 0.225 * | 0.260 * | 0.030 | 0.259 * | 0.266 * | −0.003 |
| Independent variable | CSR Investment | 0.447 *** | 0.264 * | 0.367 ** | 0.408 *** | 0.253 * | 0.444 *** |
| F value | | | 5.638 *** | 3.560 ** | 7.529 *** | 4.486 ** | 3.879 ** |
| $R^2$ | | | 0.227 | 0.156 | 0.315 | 0.215 | 0.191 |
| Modified $R^2$ | | | 0.187 | 0.112 | 0.273 | 0.167 | 0.142 |

The regression coefficients are standardized. * $p < 0.05$, ** $p < 0.01$, *** $p < 0.001$.

As for t + 1 year, CSR investment was a significant variable in ROA (β = 0.408, $p < 0.001$), ROE (β = 0.253, $p < 0.05$), and ROS (β = 0.444, $p < 0.001$) models. F value of ROA model is 7.529 ($p < 0.001$) and the modified $R^2$ was 0.273. As for ROE model, F value was 4.486 ($p < 0.01$), and the modified $R^2$ was 0.167. F value of ROS model was 3.879 ($p < 0.01$), and the adjusted $R^2$ was 0.142.

In recent years, the relationship between ESG evaluation and business performance has been the central issue in this field. The current study developed ESG indices through text mining. In addition, we divided the CSR investment into high and low groups for comparative analysis and confirmed that the low group achieved positive and short-term performance in CSR investment.

## 5. Discussion and Future Research

### 5.1. Conclusions

The topic of this study is mainly about a recently emerging ESG. ESG index was developed using the ESG keyword of the researcher's previous study result extracted from the Sustainability Report. ESG index of 49 companies was calculated by year through three methodologies extracted by the NMF technique and based on the average value of each index, 17 companies with an average value or higher were divided into high groups and 14 companies with an average value or lower were analyzed empirically. As a result of regression analysis, the relationship between CSR investment and management performance was not significant with the high group, but the CSR investment and management performance of the low group had a significantly positive (+) effect on both t and t + 1.

From a non-financial point of view, ESG management is based on environmental protection by creating an eco-friendly ecosystem, continuous management of supply chains, global social contribution, and establishment of a diverse and healthy corporate culture. On the other hand, CSR investment refers to the amount of expenditure used to solve problems in the community and improve the quality of life through activities included in social contribution during ESG management. Among them, donations are material contribution activities through direct donations and correspond to charitable donations, so it was confirmed that social contribution activities through direct donations are dominant in the case of low groups. In addition, with the descriptive statistics, the average CSR investment of the high group was 0.105, and the value of the low group was 0.151, indicating that the low group had higher CSR investment than the high group. The low group had a significantly positive (+) effect on CSR investment and corporate management performance because the companies in the low group are more active in CSR investment and tend to focus more on short-term profits. On the other hand, it can be seen that the advertising ratio is high at 0.577 and the low group at 0.012. This shows that the companies in the high group are gradually investing in advertisements rather than CSR investments to appeal to various stakeholders as well as advertisements for products. As such, low groups focus more on ESG from a financial perspective than ESG from a non-financial perspective. Various stakeholders will recognize the importance of mid-to-long-term strategies based on sustainable management rather than short-term performance and practice ESG management from a non-financial perspective.

### 5.2. Discussion of Implications

The implications of the current study are as below: First, the prior studies mainly used KEJI, ESG, and K-GWP, which are indicators to evaluate sustainability. This study is meaningful in that we developed ESG indexes based on ESG keywords extracted using text mining. Second, a quantitative ESG index was developed by extracting weights through TF-IDF techniques and multiplying the frequency of inverse documents. This provides the basis for generalizing the above three indices. Third, by quantifying the ESG index and comparing the sample companies into high and low groups, the relationship between CSR investment and corporate performance could be confirmed. As a result, companies from low group focused more on ESG from a financial perspective than ESG from a non-financial perspective. Fourth, by dividing a company's sustainability into financial and non-financial perspectives, basic data was presented to seek efficient management measures that companies should consider when establishing strategies for ESG. Finally, this study presented a new perspective on corporate ESG evaluation and meaningful exploratory implications by analyzing the relationship between CSR investment and corporate management performance.

### 5.3. Limitations and Future Research

This study examined the company's ESG management from a financial and non-financial perspective. Unlike previous studies, developing the ESG index through non-quantitative data can be meaningful, but limitations exist in sample data and measurement.

First, 339 data from 49 companies that published Sustainable Reports from 2013 to 2020 were used, but the number of samples is quite limited. If it becomes mandatory to publish reports for all companies, more data would be available for researchers. Second, it is necessary to consider both financial and non-financial performance in the impact of CSR investment on corporate management performance. It is necessary to analyze whether the ESG index and CSR investment have a significant effect on Tobin's Q to confirm the relationship with corporate value, and to study the relationship with ESG by reorganizing key performance indicators inside and outside the company into four perspectives using the balance scorecard (BSC).

Third, we used the size of the company as a control variable in analyzing the relationship between CSR investment and corporate management performance. If we considered the company's size when developing the ESG index, the index could be much more generalized. In addition, many companies are investing in advertising expenses to raise the company's image among stakeholders, so it would be meaningful to conduct further research from the perspective of advertising expenses.

Fourth, rather than focusing on CSR investment, it is necessary to check whether it affects a company's management performance through R&D expenses when analyzing high group companies. For example, CSR investment invests in planting trees in the automobile industry, but ESG management invests in R&D expenses to develop hybrid, hydrogen, and electric vehicles, so it is necessary to analyze the relationship with R&D expenses. Lastly, it will help improve the ESG index if various stakeholders' opinions on sustainable management are collected, and each company's focus in ESG management is considered. It will be more meaningful as a reliable ESG index for corporate practitioners from a strategic point of view. Further studies on ESG management should be continued to improve ESG in the business field, not only for large companies, but also for small and medium-sized companies and ventures.

**Author Contributions:** Conceptualization, J.Y. and J.L.; methodology, J.L.; software, J.L.; validation, J.L.; formal analysis, J.Y.; investigation, J.L.; resources, J.L.; data curation, J.L.; writing-original draft preparation, J.Y.; writing-review and editing, J.L.; visualization, J.L.; supervision, J.L.; project administration, J.L.; funding acquisition, J.Y. and J.L. All authors have read and agreed to the published version of the manuscript.

**Funding:** This research received no external funding.

**Institutional Review Board Statement:** Not applicable.

**Informed Consent Statement:** Not applicable.

**Data Availability Statement:** This research used the sustainability reports which are freely available from each company's webstie.

**Conflicts of Interest:** The authors declare no conflict of interest.

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
