# Peer review of "Analysis of the Relationship between Corporate CSR Investment and Business Performance Using ESG Index—The Use-Case of Korean Companies"

_sustainability, doi:10.3390/su14052911_

Round 1

Reviewer 1 Report

Well done. It is very interesting research paper. Worth to publish.

Author Response

I checked typos and improved the paper a bit. 

Reviewer 2 Report

Manuscript ID: sustainability-1601997

Title: "Analysis of the relationship between corporate CSR investment and business performance using ESG index - Focused on Korean companies"

FIRST REVIEW REPORT

In the manuscript, the Authors developed an index for ESG targeting companies that issue sustainability reports.

The topic of the paper is interesting as well as the potential academic contribution of the work. It is necessary to improve the research according to the following indications to make it suitable for the journal.

1. In the introduction section:

1.1. The Authors should discuss international situation, regulations, and approaches, and should motivate their research to be of high interest for the addressees.

1.2. It should be explained how the article has been structured by presenting the different sections.

2. Literature review should be expanded. References are not sufficient. The following studied could be considered:

https://doi.org/10.3390/su14020843

https://doi.org/10.3390/su12166387

https://doi.org/10.1016/j.jclepro.2021.128232

3. It should be discussed how the results can be interpreted in perspective of previous studies and working hypotheses.

4. A conclusion section should be inserted.

5. Tables and figures should report the sources.

6. An extensive editing of English language and style is required.

Reviewer 3 Report

Thank you for allowing me to review this very interesting paper. I think that it is quite well written and that it deserves to be published.

However there are some aspects that should be improved.

It would be good if the authors could highlight from the introduction how their paper adds value to any existing theory.

"e business 9th May 46
2019 "climate commitment" was the subject 2020 years on clean energy-related businesses 47
to implement climate pledges month, 20 billion. It" ok, but these facts are not generally known, so please cite them.

These aspects need to be referenced as they are not general knowledge.

"n addition, it decided to use 100% 49
renewable energy by 2030 and signed a contract in June 2021 to procure eco-friendly en- 50
ergy at 14 solar and wind power plants in the U.S. and Europe. It is also working with all 51
its partners to achieve its goal of "zero" carbon emissions by 2"

The aspects with Amazon must also be cited.

"POSCO, which p" please cite that.

"For the first time in Korea, in 63
December 2020, the company declared "2050 Carbon Neutral,"" please cite that

"ocial contribution in ESG management refers to corporate charity activities and the 74
Global Reporting Initiative (GRI) standard specifies the performance of social contribution 75
activities as "GRI413-Local Comm" from where do you know that? Please cite it.

"According to an analysis of 220 out of the 80
top 500 companies based on sales in 2019, social contribution expenditure was 2.99271.1 81
trillion won, and the average expenditure of one company was 13.6351 billion won, up 82
14.8% from the previous year. " Who conducted the analysis? Can you include this information in your paper?

The introduction must be clearer, i.e. you need to explain:

  • the research gap
  • the research question
  • the theory on which the paper is based
  • how you implement the research question in the paper, especially in the methodology
  • what is the originality of your paper
  • the last paragraph of the introduction should contain a brief description of the next sections of your paper

Lit review. More references on the topic should be cited. 

Table 1: I do not understand how the table was compiled. From where are these information taken? What are the references for the table? How was the rating assessed? What is MSCI? What is KCGS? How are these aspects rated-evaluated? Refinitv. What is that? When are 0 points given and when are 9 given? Please explain!

Is the table your own compilation it is the data taken from somewhere else?

The relationship between corporate CSR investment and business performance in terms of sustainable organizational performance, sustainable product lifecycle management, and sustainable Industry 4.0 has not been covered, and thus such recent references should be cited: Kovacova, M., and Lăzăroiu, G. (2021). “Sustainable Organizational Performance, Cyber-Physical Production Networks, and Deep Learning-assisted Smart Process Planning in Industry 4.0-based Manufacturing Systems,” Economics, Management, and Financial Markets 16(3): 41–54. doi: 10.22381/emfm16320212. Dawson, A. (2021). “Robotic Wireless Sensor Networks, Big Data-driven Decision-Making Processes, and Cyber-Physical System-based Real-Time Monitoring in Sustainable Product Lifecycle Management,” Economics, Management, and Financial Markets 16(2): 95–105. doi: 10.22381/emfm16220216. Nica, E., and Stehel, V. (2021). “Internet of Things Sensing Networks, Artificial Intelligence-based Decision-Making Algorithms, and Real-Time Process Monitoring in Sustainable Industry 4.0,” Journal of Self-Governance and Management Economics 9(3): 35–47. doi: 10.22381/jsme9320213. The relationship between corporate CSR investment and business performance in terms of sustainable smart manufacturing, sustainable industrial value creation, and sustainable cyber-physical production systems has not been covered, and thus such recent references should be cited: Popescu, G. H., Petreanu, S., Alexandru, B., and Corpodean, H. (2021). “Internet of Things-based Real-Time Production Logistics, Cyber-Physical Process Monitoring Systems, and Industrial Artificial Intelligence in Sustainable Smart Manufacturing,” Journal of Self-Governance and Management Economics 9(2): 52–62. doi: 10.22381/jsme9220215. Nica, E., Stan, C. I., LuÈ›an (Petre), A. G., and OaÈ™a (Geambazi), R.-Ș. (2021). “Internet of Things-based Real-Time Production Logistics, Sustainable Industrial Value Creation, and Artificial Intelligence-driven Big Data Analytics in Cyber-Physical Smart Manufacturing Systems,” Economics, Management, and Financial Markets 16(1): 52–62. doi: 10.22381/emfm16120215. Lăzăroiu, G., Kliestik, T., and Novak, A. (2021). “Internet of Things Smart Devices, Industrial Artificial Intelligence, and Real-Time Sensor Networks in Sustainable Cyber-Physical Production Systems,” Journal of Self-Governance and Management Economics 9(1): 20–30. doi: 10.22381/jsme9120212.   As section 2 is a literature review you need to cite references, this means that in each paragraph there should be at least one scientific reference. Figure 1 and 2. From where is that information? Any references for that?   Research methodology. More information and more references need to be cited. It is not clear how this framework was implemented-developed. Table 2. Descriptive statistics and correlation of highly clustered variables ... it is not clear how the data was measured. Beneath each table you need to add a note and explain the abbreviations that you have used. Discussion means that you compare own results with previous findings from the literature. I do not see here any comparisons between own results and previous results from the literature. This section must also contain a critical assessments of your findings.   You need to have conclusions. Conclusions must contain: theoretical contributions; managerial implications; limitations; future research perspectives.   More up to date references must be added 

Round 2

Reviewer 2 Report

The Authors improved their manuscript according to the suggestions of my previous review report.

Now the paper is suitable for the journal.

Congratulations!

Reviewer 3 Report

Thank you for implementing all suggestions and recommendations. Back luck in attracting lots of citations